# Automatic Concept Extraction for Concept Bottleneck-based Video Classification

## Abstract

Recent efforts in interpretable deep learning models have shown that concept-based explanation methods achieve competitive accuracy with standard end-to-end models and enable reasoning and intervention about extracted high-level visual concepts from images, e.g., identifying the wing color and beak length for bird-species classification. However, these concept bottleneck models rely on a domain expert providing a necessary and sufficient set of concepts–which is intractable for complex tasks such as video classification. For complex tasks, the labels and the relationship between visual elements span many frames, e.g., identifying a bird flying or catching prey–necessitating concepts with various levels of abstraction. To this end, we present CoDEx, an automatic Concept Discovery and Extraction module that rigorously composes a necessary and sufficient set of concept abstractions for concept-based video classification. *CoDEx* identifies a rich set of complex concept abstractions from natural language explanations of videos–obviating the need to predefine the amorphous set of concepts. To demonstrate our method's viability, we construct two new public datasets that combine existing complex video classification datasets with short, crowd-sourced natural language explanations for their labels. Our method elicits inherent complex concept abstractions in natural language to generalize concept-bottleneck methods to complex tasks.

## 1 Introduction

Deep neural networks (DNNs) provide unparalleled performance when applied to application domains, including video classification and activity recognition. However, the inherent black-box nature of the DNNs inhibits the ability to explain the output decisions of a model. While opaque decision-making may be sufficient for certain tasks, several critical and sensitive applications force model developers to face a dilemma between selecting the best-performing solution or one that is inherently explainable. For example, in the healthcare domain (Yeung et al. (2019)), a life-or-death diagnosis compels the use of the best performing model, yet accepting an automated prediction without justification is wholly insufficient. Ideally, one could take advantage of the power of deep learning while still providing a sufficient understanding of why a model is making a particular decision, especially if the situation demands trust in a decision that can have severe impacts.

To address the need for model interpretability, researchers have sought to enable model intervention by leveraging concept bottleneck-based explanations. Unlike post hoc explanation methods–where techniques are used to extract an explanation for a given input for an inference by a trained black-box model (Chakraborty et al. (2017); Jeyakumar et al. (2020)), concept bottleneck models are inherently interpretable and take a human reasoning-inspired approach to explaining a model inference based on an underlying set of concepts that define the decisions within an application. Thus far, prior works have focused on concept-based explanation models for image (Kumar et al. (2009); Koh et al. (2020)) and text classification (Murty et al. (2020)). However, the concepts are assumed to be given a priori by a domain expert–a process that may not result in a necessary and sufficient set of concepts. For instance, for bird species identification, an expert may provide two redundant concepts that are possibly correlated, such as wing color and beak color. More critically, prior works have considered simple concepts with the same level of abstraction, e.g., visual elements present in a single image. For more complex tasks such as video activity classification, a label may span multiple frames. Thus, the composing set of concepts will have various levels of abstraction representing relationships of various

visual elements spanning multiple frames, e.g., a bird flapping its wings. Unlike the prior works, we aim to exploit the complex abstractions inherent in natural language explanations to conceptualize such complex events.

**Research Questions.** In summary, this paper seeks to answer the following research questions:

- How can a machine automatically elicit the inherent complex concepts from natural language to construct a necessary and sufficient set of concepts for video classification tasks?

- Given that a machine can extract such concepts, are they informative and meaningful enough to be detected in videos by DNNs for downstream prediction tasks?

- Are the machine extracted concepts perceived by humans as good explanations for the correct classifications?

**Approach.** This paper introduces an automatic concept extraction module for concept bottleneck-based video classification. The bottleneck architecture equips a standard video classification model with an intermediate concept prediction layer that identifies concepts spanning multiple video frames. To compose the concepts that will be predicted by the model, we propose a natural language processing (NLP) based automatic Concept Discovery and Extraction module, CoDEx, to extract a rich set of concepts from natural language explanations of a video classification. NLP tools are leveraged to elicit inherent complex concept abstractions in natural language. CoDEx identifies and groups short textual fragments relating to events, thereby capturing the complex concepts from videos. Thus, we amortize the effort required to define and label the necessary and sufficient set of concepts. Moreover, we employ an attention mechanism to highlight and quantify which concepts are most important for a given decision.

To demonstrate the efficacy of our approach, we construct two new datasets–MLB V2E (Video to Explanations) for baseball activity classification and MSR-V2E for video category classification–that combine complex video classification datasets with short, crowd-sourced natural language explanations for their corresponding labels. We first compare our model against the existing standard end-to-end deep-learning methods for video classification and show that our architecture provides additional benefits of an inherently interpretable model with a marginal impact on performance (less than 0.3% accuracy loss on classification tasks). A subsequent user study showed that the extracted concepts were perceived by humans as good explanations for the classification on both the MLB-V2E and MSR-V2E datasets.

**Contributions.** We summarize our contributions as follows.

- We propose *CoDEx*, a concept discovery and extraction module that leverages NLP techniques to automatically extract complex concept abstractions from crowd-sourced, natural language explanations for a given video and label–obviating the need to manually define a necessary and sufficient set of concepts.

- We evaluate our approach on complex video classification datasets and show that our model attains high concept accuracies while maintaining competitive task performance with standard end-to-end video classification models.

- We also augment the concept-based explanation architecture to include an attention mechanism that highlights the importance of each concept for a given decision. We show that users prefer our concept extraction method over baseline methods to explain a given label.

- We construct two new public datasets, MLB-V2E and MSR-V2E, that combine complex video classification datasets with short, crowd-sourced natural language explanations and labels.

## 2 RELATED WORK

There is a wide array of works in explainable deep learning for various applications. This work focuses on the concepts-based explanations for video classification, and this section provides an overview of the existing literature for overlapping domains.

**Concept-Based Explanations for Images and Text.** A number of existing works consider concept-bottleneck architectures where models are trained to interact with high-level concepts. Generally, the

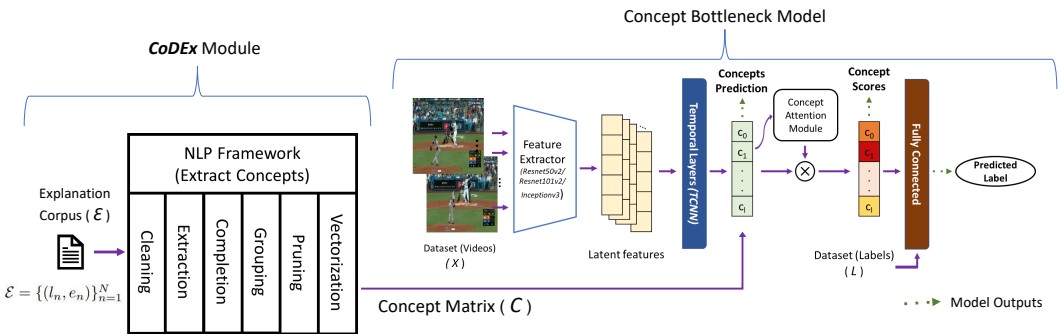

Figure 1: The overall pipeline showing the automatic concept extraction framework from natural language explanations and the concept bottleneck classification model training framework.

approaches are multi-task architectures, where the model first identifies a human-understandable set of concepts and then reasons about the identified concepts. Until now, the applications have been limited to static image and text applications. Koh et al. (2020) used pre-labeled concepts provided by the dataset to train a model that predicts the concepts, which is then used to predict the final classification. However, the caveat is that the concepts had to be manually provided. Ghorbani et al. (2019) and Yeh et al. (2020) proposed approaches that automatically extract groups of pixels from the input image that represent meaningful concepts for the prediction. They were designed largely for image classification and extract concepts directly from the dataset. Kim et al. (2018) propose a post-hoc explanation method that returns the importance of user-defined concepts for a classification. In the mentioned works, the concepts have been limited to simple concepts and are not suited for complex tasks such as video classification where we have complex concepts that may span multiple frames with various levels of abstraction.

**Explanations for Video Classification** Other approaches have been considered to explain video classification and activity recognition. Chattopadhay et al. (2018) applied GradCAM and GradCAM++ to video classification, where for each frame, the important region of the frame to the model is highlighted as a heatmap. Hiley et al. (2020) extract both spatial and temporal explanations from input videos by highlighting the relevant pixels. However, these are post-hoc techniques that focus on explaining black-box models, whereas our approach enables concept-bottleneck methods for video classification that are intended to be inherently interpretable and intervenable.

**Video Captioning.** In recent years, there is a large number of works (Pan et al. (2017); Gao et al. (2017); Wang et al. (2018); Yan et al. (2019); Zhou et al. (2018); Chen & Jiang (2021); Yu et al. (2017)) on video captioning. While they also employ natural language techniques, these works are tangential to generating text explanations for classifications, since they are merely describing the video. Our model provides an explanation justifying the *classification* of the video. Similarly, the associated datasets such as MSR-VTT (Xu et al. (2016)) only have videos with ground truth captions that only describe the video without the context of a classification–which often results in concepts that do not pertain to a classification.

**Semantic Concept Video Classification.** The closest works to this paper is the body of work in semantic concept video classification (Fan et al. (2004; 2007)), where the concepts are defined as salient objects that are visually distinguishable video components. The concepts in these works are simple objects detected in the videos and are not complex enough to capture the semantics of events that happen over multiple frames of the videos. These works typically used traditional SVM-based video classifiers. Assari et al. (2014) represent a video category based on the co-occurrences of the semantic concepts and classify based on the co-occurrences, but their method requires a predefined set of concepts. Thus, we now present the methodology behind our automatic concept extraction for concept bottleneck video classification.

# 3 CONCEPT DISCOVERY AND BOTTLENECK VIDEO CLASSIFICATION

This work introduces *CoDEx*, an automatic concept extraction method from natural language explanations for concept-based video classification. Figure 1 depicts the overall concept-bottleneck pipeline, composed of *CoDEx* and the concept bottleneck model. *CoDEx* extracts a set of concepts

from natural language explanations that will comprise the bottleneck layer for the video classification model. We first formalize the overall problem and then provide the methodology for both modules.

**Problem Formalization.** We assume that we have a training dataset $\{(x_n, l_n)\}_{n=1}^N = \mathcal{D}$ of videos $x_n$ with a label $l_n \in \mathcal{L}$, where $\mathcal{L}$ is a predefined set of possible class labels for the video. Each video is represented as a sequence of frames $f \in \mathcal{F}$ where $\mathcal{F}$ is the set of video frames. Thus video $x_n = \langle f_{n0}, f_{n1}, \dots, f_{nT} \rangle$, where $f_{nt}$ represents frame $t$ of video $n$. For each video $x_n$, we form a label-explanation pair $(l_n, e_n)$, where $e_n$ is a (short) natural language document explaining the given label $l_n$. If multiple annotators contribute to an explanation for video-label pair, $(x_n, l_n)$, then these are concatenated to form $e_n$. The full set of pairs $\mathcal{E} = \{(l_n, e_n)\}_{n=1}^N$ is the *explanation corpus*. Thus, the design goals are:

- **Concept Discovery and Extraction (CoDEx) Module:** Given the explanation corpus, first produce an $N \times K$ concept matrix, $C$, where the $(n, k)$th element is 1 if the $n$th explanation contains discovered concept $k$ and 0 otherwise. We call the $n$th row $\mathbf{c}_n$, the concept vector for video $x_n$. $K$ is the total number of discovered concepts.

- **Concept Bottleneck Model:** Given a concept matrix, $C$, the second goal is to train a concept bottleneck model such that for a given video $x_i$, we predict a concept vector $\mathbf{c}_i-$ which indicates the presence or absence of concepts and their importance scores. The model then makes use of $\mathbf{c}_i$ to make the final video classification.

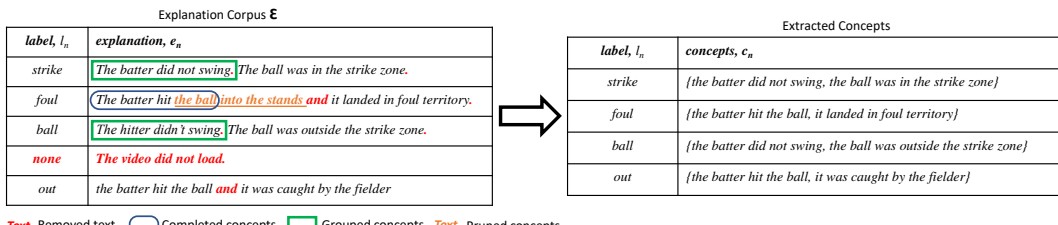

Figure 2: Running example for all six stages of the discovery pipeline module. The left table is the explanation corpus, with highlighted fragments to be modified. The right table contains the discovered concepts. The detailed step-by-step modifications are provided in Appendix A.1.

### 3.1 CoDEx: Concept Discovery and Extraction module

We now describe *CoDEx*, that extracts concepts from the explanation corpus, $\mathcal{E}$. The automatic extraction of the significant concepts is done in 6 steps, as outlined in Fig. 1. These are: **cleaning**, **extraction**, **grouping**, **completion**, **pruning**, and **vectorization**, which produce the final concept matrix, $C$. Each of these steps are described below and illustrated with an example corpus depicted in Figure 2.

**Cleaning.** We remove explanations associated with corrupted or unlabeled videos from the explanation corpus. In Figure 2, this phase would remove the fourth row with the "none" label.

**Extraction.** The objective of this phase is to identify sentence constituents relevant to explaining the label. These text fragments, short sequences of words that appear in the document, are referred to as *raw concepts*. To achieve this, the cleaned explanation corpus is tokenized then passed through a pretrained constituency parser to recursively decompose the sentences. At each level of the constituency hierarchy, the text fragments are evaluated to determine whether they constitute a candidate raw concept. The rules for candidate raw concepts include the inclusion and exclusion rules below and follow the widely adopted Universal POS tag naming convention for token types (Petrov et al. (2012)). Every constituency parsed phrase that satisfies one of the two inclusion rules and not the exclusion rule is considered a candidate concept.

| rule name | rule |
|---|---|
| Inclusion 1. | noun/pronoun → auxillary (optional) → particle (optional) → verb (optional) |
| Inclusion 2. | noun/pronoun → auxiliary whose lemma is 'be' → any token |
| Exclusion | subordinating conjunction |

Table 1: Inclusion and exclusion rules for candidate concepts.

After the extraction process is completed, we have a set of raw concepts, $\widetilde{\mathcal{K}}$, and each video is associated with a subset of these raw concepts. An example of extracted raw concepts, $\widetilde{\mathcal{K}}$, can be found in Appendix A.1.

**Completion.** There are instances where the pretrained constituency parser will split sentences midway through a text fragment in one sentence that was kept whole in another. For instance, in Figure 2, the constituency parser splits the explanation for "foul" such that "the batter hit the ball" is incorrectly excluded from the raw concepts. To ensure that those concepts are captured, we perform a substring lookup of each raw concept through all documents of the explanation corpus and count an explanation as containing a raw concept if it contains the corresponding raw concept as a substring. This does not change the number of raw concepts identified but increases their frequency counts.

**Grouping (similar raw concepts).** When identical text fragments are identified in different explanations, they are counted directly as the same *raw concept*. However, we would ideally like to treat superficially different concepts as the same if they essentially carry the same meaning, e.g., Figure 2 highlights two different raw concepts that carry the same meaning and hence can be grouped. For this, we use agglomerative clustering (Müllner (2011)) approach that measures the degree of difference between pairs of raw concepts and groups them together if they are similar enough. Our key contribution here is the *distance metric* used in clustering which is a novel measure of meta-distance between raw concepts. This measures the difference between concepts based on two aspects of the raw concepts: their linguistic difference and their difference in terms of the label categories with which they are associated.

We define meta-metric, $d$, as combining a linguistic distance, $d_{\text{text}}$ (capturing linguistic difference) as well as a meta-metric, $d_{\text{label}}$ (capturing the difference in the labels associated with each raw concept). More formally, for two raw concepts $\kappa_i, \kappa_j \in \widetilde{\mathcal{K}}$ our distance is linear combination:

$$d(\kappa_i, \kappa_j) = d_{\text{text}}(\mathbf{v}_i, \mathbf{v}_j) + \lambda d_{\text{label}}(\mathbf{n}_i, \mathbf{n}_j) \tag{1}$$

where $\mathbf{v}_i$ is a sentence embedding for the text fragment of concept $\kappa_i$ (e.g., based on the BERT model Devlin et al. (2019)), $d_{\text{text}}$ is a standard distance measure between vectors (e.g., cosine distance), $d_{\text{label}}$ is a meta-distance which aims to capture the similarity between two label count vectors, and $\lambda$ is a hyperparameter controlling the relative importance between textual and label distance. The inclusion of a label distance helps to distinguish between concepts that are superficially linguistically very simlar, but have very distinct meanings within the domain of interest. For instance, without the $d_{\text{label}}$, the concepts "the ball passed inside the strike zone" and "the ball passed outside the strike zone" will be grouped together though they are very different concepts as they have a very small $d_{\text{text}}$. We provide a more formal definition of the meta-metric $d_{\text{label}}$ more formally and provide some intuition behind its construction in Appendix A.3. We also exclude concept groups which occur very rarely in the explanation corpus, with frequency less than some small threshold, $t$.

**Pruning.** Here, we seek a compact subset of concepts that, together, capture a high degree of information about the label while maintaining interpretability. More formally, after grouping, we have a set of raw concepts $\widetilde{\mathcal{K}} = \{\kappa_1, \ldots, \kappa_J\}$, and we seek some subset of maximally informative concepts $\mathcal{K}^\star = \{\kappa_{j_1} \ldots, \kappa_{j_K}\} \subseteq \widetilde{\mathcal{K}}$.

To see what is meant by *maximally informative*, consider a randomly selected entry in the explanation corpus $(l, e)$. We define a binary random variable, $\mathrm{C}_j$ for each raw concept $\kappa_j$, and for any concept set $\mathcal{K} = \{\kappa_{j_1}, \ldots, \kappa_{j_K}\}$, random vector $\mathrm{C}_{\mathcal{K}} = [\mathrm{C}_1, \ldots, \mathrm{C}_K]$, such that $\mathrm{C}_j = 1$ if $\kappa_j \in e$ and $0$ otherwise. $Y$ is the random variable which takes label $l$. We wish to choose the smallest subset of concepts such that the mutual information (MI)[1] between chosen concepts, $\mathcal{K}$, and label, $Y$, given by $I(Y; \mathrm{C}_{\mathcal{K}})$, is greater than a threshold fraction, $\gamma < 1$ of the MI between the label and the complete concept vector, $I(Y; \mathrm{C}_{\widetilde{\mathcal{K}}})$. That is to say we wish to find $\mathcal{K}$ which satisfies:

$$I(Y; \mathrm{C}_{\mathcal{K}}) \geq \gamma I(Y; \mathrm{C}_{\widetilde{\mathcal{K}}}) \tag{2}$$

and where there is no subset $\mathcal{K}' \subseteq \widetilde{\mathcal{K}}$ with $|\mathcal{K}'| < |\mathcal{K}|$ which also satisfies Equation equation 2. In practice, this is infeasible as the problem is combinatorial. However, we note that $f(\mathcal{K}) = I(Y; \mathrm{C}_{\mathcal{K}})$ is a monotone submodular set function of $\widetilde{\mathcal{K}}$. Given this, if we recursively construct a set of size $K$, by greedily adding single concepts that most improve the MI, the resulting set will be at least $1 - \frac{1}{e}$

---

[1] We use the standard definition of mutual information (MI) for discrete random variables (MacKay (2003)).

as good as the most informative set of size $K$ (Nemhauser et al. (1978)). Therefore, we guarantee a highly-informative set $\mathcal{K}^\star$ by iteratively adding concepts to those previously selected, greedy with respect to the MI, until we have a set that satisfies Equation 2.

**Vectorization.** Each concept $\kappa_{j_k} \in \mathcal{K}^\star$ is given a unique index $k \in \{1, \ldots, K\}$, and each data-point, $x_n$ is associated with a concept vector $\mathbf{c}_n = (c_{n1}, \ldots, c_{nK})$, where $c_{nk} = 1$ if $\kappa_{j_k} \in e_n$ and 0 otherwise, indicating the presence or absence of the $k$th concept in the $n$th explanation. The collection of all the concept vectors gives an $N \times K$ concept matrix, $\mathbf{C}$.

## 3.2 Concept Bottleneck Model

We use the videos, the extracted concepts from *CoDEx*, and the labels to train an interpretable concept-bottleneck model to predict the activity and the corresponding concepts. Figure 1 shows the overview of our bottleneck architecture. The activity label, the concepts, and the corresponding concept scores are the outputs of the interpretable model and are indicated by dotted arrows in Figure 1.

Our bottleneck model architecture is based on the standard end-to-end video classification models where we use convolutional neural network-based feature extractors pretrained on the Imagenet dataset Deng et al. (2009) to extract the spatial features from the videos. The features are then passed through temporal layers that can capture features across multiple frames which in turn is bottle-necked to predict the concepts. Lastly, we deploy an additive attention module (Bahdanau et al. (2014) that gives the concept score $\alpha_c$ indicating the importance of every concept to the classification. The attention module also improves the interpretability of the the bottleneck model by indicating the key concepts for classification and this is evaluated in section 5. More details regarding the model architecture and hyper-parameters are in the Appendix A.5

**Model loss function.** The entire bottleneck classification model is trained in an end-to-end manner. Since the concepts are represented as binary vectors, we use sigmoid activation on the concept bottleneck layer and binary categorical loss function as the concept loss. The final layer of the classifier has softmax activations and categorical cross-entropy as the classification loss function. Thus, the overall loss function of the model is the sum of concept loss and the classification loss. The hyperparameter $\beta$ controls the tradeoff between concept loss, $L_C$, versus classification loss, $L_Y$ as shown in equation 3. The full expansion of the equation is located in Appendix A.5.

$$Loss(L) = \frac{1}{N} \sum_{n=1}^{N} \left( L_{Y_n} + \beta \times L_{C_n} \right) \qquad \text{where} \qquad \beta > 0 \tag{3}$$

**Testing phase.** Given an input test video, the model provides us with the activity prediction (label of the video), a concept vector indicating the relevant concepts that induced this classification and the concept importance score for each concept. By retrieving the phrase representing the concepts present in the video, the result obtained is a human-understandable explanation of the classification.

## 4 Implementation

To demonstrate our automatic concept extraction method, we construct two new datasets - MLB-V2E (Video to Explanations) and MSR-V2E, which combines short video clips with crowd-sourced classification labels and corresponding natural language explanations. For both datasets, we obtained a video activity label and natural language explanations for that label by crowd-sourcing on Amazon Mechanical Turk and used unrestricted text explanations to extract concepts automatically. For IRB exemption and compensation information, please refer to the Ethics Statement.

**MLB-V2E Dataset:** We used a subset of the MLB-Youtube video activity classification dataset introduced by Piergiovanni & Ryoo (2018)–which had segmented video clips containing the five primary activities in baseball: strike, ball, foul, out, in play. We preprocessed the dataset and extracted 2000 segmented video clips where each video was 12 seconds long, 224×224 in resolution, and recorded at 30 fps. To ensure that the quality of explanations is good, we screened over 450 participants. Based on their baseball knowledge, 150 participants were qualified to provide the natural language text explanations for our video clips. We have included a sample of our screening survey, the primary survey, and the explanations collected in the supplementary materials.

| Dataset | Number of Concepts after each Phase | | | |
|---|---|---|---|---|
| | Extraction | Completion | Grouping | Pruning |
| MLB-V2E | 1885 | 1885 | 225 | 80 |
| MSR-V2E | 1678 | 1678 | 104 | 62 |

Table 2: The number of concepts extracted by the Concept Discovery module from the explanation corpus after every phase.

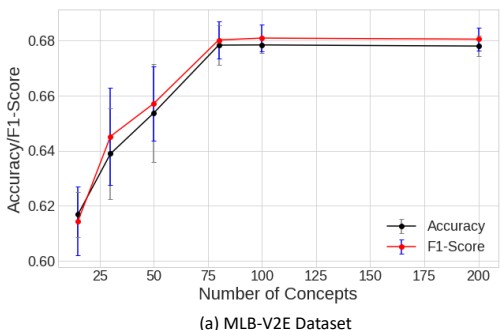
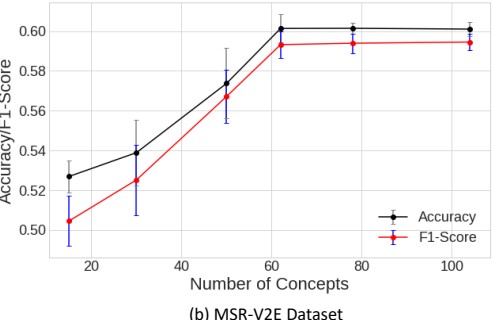

(a) MLB-V2E Dataset          (b) MSR-V2E Dataset

Figure 3: The number of concepts versus performance trade-off for the (a) the MLB-V2E dataset and (b) the MSR-V2E dataset.

**MSR-V2E Dataset:** For this dataset, we used 2020 video clips from the MSR VTT dataset introduced by Xu et al. (2016). The MSR-VTT dataset has general videos from everyday life and descriptions of these videos associated with them. Each video clip is between 10-30 seconds long, and approximately 200 participants provided the labels and explanations to construct the MSR-V2E dataset. The videos are classified into ten categories: Automobiles, Cooking, Beauty and Fashion, News, Science and Technology, Eating, Playing Sports, Music, Animals, and Family (more details in Appendix A.11). **Training:** All our models were trained on $2 \times$ Titan GTX GPUs using Adam optimizer. A summary of our entire model architecture and a trained model is provided in the supplementary materials.

## 5 RESULTS

**Number of extracted concepts.** Table 2 shows that the system extracted 80 concepts and 62 concepts from the explanation corpus of MLB-V2E and MSR-V2E respectively. The number of concepts remaining after the pruning phase is determined by the cumulative Mutual Information(MI) threshold. To identify the best threshold, we plotted the number of concepts at different thresholds versus performance of the model as shown in Figure 3. We found that the task classification performance did not increase after a certain number of concepts and that optimal spot for the number of concepts corresponded to 90% Mutual Information (Appendix A.6).

**Comparing concept-bottleneck models to baselines.** We adopt model architectures and hyperparameters from standard well-performing approaches that fall under 3 categories: 1) without concept bottleneck, 2) with concept bottleneck, 3) with concept bottleneck and attention. We compared the performance of models with the bottleneck layer with standard video classification models without a concept bottleneck layer. We find that, though the latent space was constrained to the limited set of concepts extracted from the explanation corpus, concept models performed as well as the uncon-

| Dataset | Feature Extractor | Model Type | Task Classification | | Concepts |
|---|---|---|---|---|---|
| | | | Accuracy(%) | F1-score | AUC |
| MLB-V2E | Inception V3 | Standard | $68.46 \pm 1.27$ | $0.68 \pm 0.011$ | - |
| | | Bottleneck | $68.16 \pm 1.12$ | $0.68 \pm 0.004$ | $0.85 \pm 0.003$ |
| | | Bottleneck + Attn. | $68.38 \pm 1.34$ | $0.68 \pm 0.004$ | $0.88 \pm 0.001$ |
| MSR-V2E | Inception V3 | Standard | $61.79 \pm 1.42$ | $0.60 \pm 0.012$ | - |
| | | Bottleneck | $61.42 \pm 1.18$ | $0.60 \pm 0.013$ | $0.83 \pm 0.006$ |
| | | Bottleneck + Attn. | $61.68 \pm 1.23$ | $0.60 \pm 0.009$ | $0.86 \pm 0.004$ |

Table 3: Performance of Models. The full table can be found in Appendix A.7.

| Dataset | Concepts Extraction | Task Accuracy(%) | Task F1-score | Concept AUC |
|---------|---------------------|------------------|---------------|-------------|
| MLB-V2E | CoDEx | 68.38 | 0.6802 | 0.8801 |
|         | w/o Extraction | 63.47 | 0.5823 | 0.8185 |
|         | w/o Grouping | 67.80 | 0.6772 | 0.8122 |
|         | w/o Pruning | 68.36 | 0.6802 | 0.8419 |
|         | w/o Grouping and Pruning | 65.29 | 0.6526 | 0.7821 |
| MSR-V2E | CoDEx | 61.68 | 0.6010 | 0.8600 |
|         | w/o Extraction | 58.02 | 0.5214 | 0.7830 |
|         | w/o Grouping | 59.31 | 0.5745 | 0.7888 |
|         | w/o Pruning | 61.68 | 0.6010 | 0.8131 |
|         | w/o Grouping and Pruning | 54.70 | 0.5178 | 0.7467 |

Table 4: Ablation Studies: Shows the effect of each step in CoDEx on model performance

strained models, on both datasets. We also find that the addition of the attention layer improves the concept prediction of the models. Table 3 shows that concept bottleneck models achieved comparable task accuracy to standard black-box models on both tasks, despite the bottleneck constraint while achieving high concept prediction performance. Appendix A.7 shows the performance with other feature extractors.

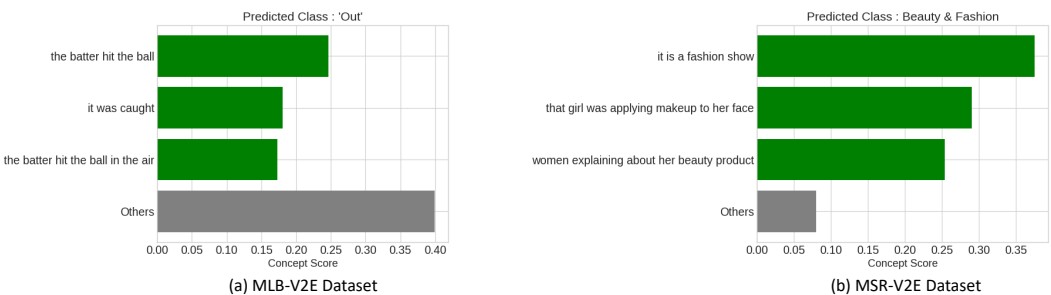

(a) MLB-V2E Dataset

(b) MSR-V2E Dataset

Figure 4: Explanation offered by the model indicating the predicted class concepts present and their corresponding scores for (a) the MLB-V2E dataset (b) the MSR-V2E dataset.

**Ablation Study of CoDEx.** We performed an ablation study to highlight the impact of CoDEx's components. For sentence-level concept extraction, we replaced the extraction phase with word-level concepts extracted from the explanation corpus. We also evaluated how removing the grouping and pruning phases would impact performance. Table 4 shows the results of our study. We find that using word-level concepts significantly reduced the performance of predicting the task and the concepts. The grouping and pruning stages had a greater impact on the concept prediction performance that affected the explainability of the model.

**Concept scores for interpretability.** Not only does the attention module increase performance in concept prediction, but it also improves the explainability of the bottleneck model by providing an importance score for the concepts. Figure 4 shows the explanation from the concept bottleneck model with attention on a test sample from the two datasets. More examples can be found in Appendix A.10. The title shows the classification label, the y-axis indicates the top-3 concepts predicted as present in the video clip, and the x-axis corresponds to the concept score. $Others$ refers to the sum of the importance of all the remaining concepts.

**Human study to evaluate concepts' explainability.** We performed a Mechanical Turk study to evaluate the explainability of our extracted complex concepts to the end-users. The participants were asked to select from four different options (presented in random) of what they consider to be the best possible Explanation for the classification of a given video. The four options are: Complex concepts predicted models without attention, Complex concepts predicted by models with attention, Concepts of a random video not belonging to the same predicted class and a Random set of 2-5 concepts from the set of the most frequent concepts. The methodology of this study was inspired by Chang et al. (2009)'s paper. Figure 5 presents the aggregated results of the Mechanical Turk study. The complex concepts predicted by the concept bottleneck model with attention was considered as the preferred explanation by 68% and 57% of the responses in the MLB-V2E and MSR-V2E datasets respectively followed by the concepts bottleneck models without attention in 20% and 28% of the responses for

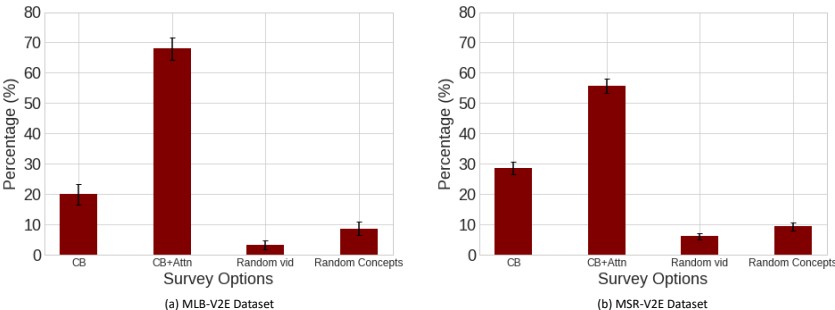

Figure 5: Survey responses with 95% bootstrap confidence interval for the two datasets

the two datasets. The presented confidence intervals are calculated using the bootstrap method as described by DiCiccio & Efron (1996) for 95% confidence.

## 6 DISCUSSION

**Annotation effort.** Although CODEX requires collecting a large explanation corpus, prior work requires two studies: one study to identify the set of essential concepts, and a second study to annotate the videos with the essential concepts–whereas CODEX only requires a single study. Further, if the vocabulary of concepts is large, the annotation process would be inconvenient for the user. Moreover, natural language explanations can express richer compositions of concepts rather than simply identifying the presence or absence of an individual concept. Thus, CODEX's annotation efforts are more expressive, less expensive and less cumbersome than prior works.

**Representative Concept.** Our concept extraction method selects the most frequent concept in a grouped cluster as the representative concept. In general, the most frequent concept suffices to explain a particular component of the complex activity. However, there were some instances where the most frequent concept would have a specific terminology rather than a general term, e.g., "left fielder" is a subclass of "outfielder." Future work can strive towards generating the representative concept for a cluster, as opposed to opting for the most frequent or popular phrasing.

**Preserving Spatial-temporal Semantics.** Our model's output explanation currently provides a set of activated concepts along with their score. However, they do not capture the spatial and temporal relationships between concepts. Some rich concepts implicitly embed spatial and temporal properties, e.g., "the batter hit the ball on the ground" implies the following sequence: a batter swung at a ball, made contact with the ball, and the ball landed on the ground. However, if the generated set of concepts is limited to less informative concepts, e.g., "the batter," the spatial and temporal ordering of concepts matters. Future work can generalize the architecture to generate concept-based natural language explanations that explicitly preserve spatial-temporal semantics.

**Neural-symbolic Reasoning.** Our model's reasoning layers are inherently black-box in nature, i.e., the concept vectors are fed into a fully connected network. To further bolster human-machine teaming and interpretability, the final classification model can be replaced with a rule-based model–analogous to prior works that fuse deep learning inferences with symbolic reasoning layers for complex event detection (Xing et al. (2020); Vilamala et al. (2019)).

## 7 CONCLUSION

The remarkable performance of deep neural networks is only limited by the stark limitation in clearly explaining their inner workings. While researchers have introduced feature highlighting explanation techniques to provide insight into these black-box models, concept-bottleneck models offer a promising new approach to explanation by decomposing application tasks into a set of underlying concepts. We build upon concept-based explanations by introducing an automatic concept extraction module, *CoDEx*, to a general concept bottleneck architecture for identifying, training, and explaining video classification tasks. In coalescing concept definitions across crowd-sourced explanations, *CoDEx* amortizes the expertise of concept definition while removing the burden from the model developer. We also show that our method provides reasonable explanations for classification without compromising performance compared to standard end-to-end video classification models.

ETHICS STATEMENT

**IRB Exemption and Compensation.** This research study has been certified as exempt from review by the IRB and the participants were compensated at a rate of 15 USD per hour for a total of 920.36 USD spent.

**Dataset privacy.** There was no personally identifiable information collected at anytime during the turk study. The responses provided by the mechanical turkers that are present in the dataset are completely anonymous.

REPRODUCIBILITY STATEMENT

The entire code with detailed comments are provided in the supplementary materials. The model architectures and hyper-parameters used are discussed in Appendix A.5. All the plots and graphs can be obtained by running the code without modifications.

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

## A  APPENDIX

### A.1  RUNNING EXAMPLE TO DESCRIBE THE CONCEPT DISCOVERY PIPELINE

**Example** (A simple Explanation corpus). *Consider using baseball domain with a few meaningful labels, a null label $\mathcal{L} = \{strike, ball, foul, out, none\}$ and five entries in the explanation corpus $\mathcal{E}$:*

| *id*, $n$ | *label*, $l_n$ | *explanation*, $e_n$ |
|---|---|---|
| 1 | strike | The batter did not swing. The ball was in the strike zone. |
| 2 | foul | the batter hit the ball into the stands and it landed in foul territory |
| 3 | ball | The hitter didn't swing. The ball was outside the strike zone. |
| 4 | none | The video did not load. |
| 5 | out | the batter hit the ball and it was caught by the fielder |

**Example** (After Cleaning Phase). *The none label entry is removed after the Cleaning phase*

| id, $n$ | label, $l_n$ | explanation, $e_n$ |
|---|---|---|
| 1 | strike | The batter did not swing. The ball was in the strike zone. |
| 2 | foul | the batter hit the ball into the stands and it landed in foul territory. |
| 3 | ball | The hitter didn't swing. The ball was outside the strike zone. |
| 4 | out | the batter hit the ball and it was caught by the fielder |

**Example** (After Extraction Phase). *The raw concepts are extracted based on the defined rules discussed in Table 1 corresponding to each explanation with the help of a pre-trained constituency parser.*

| id, $n$ | label, $l_n$ | raw concepts, $\widetilde{\mathcal{K}}$ |
|---|---|---|
| 1 | strike | The batter did not swing, The ball was in the strike zone |
| 2 | foul | the ball into the stands, it landed in foul territory |
| 3 | ball | The hitter didn't swing, The ball was outside the strike zone |
| 4 | out | the batter hit the ball, it was caught by the fielder |

**Example** (After Completion Phase). *The concept 'the batter hit the ball' was not extracted by the Extraction phase for the Id 2 explanation in this corpus. This missing concept is retrieved through the sub-string matching.*

| id, $n$ | label, $l_n$ | raw concepts, $\widetilde{\mathcal{K}}$ |
|---|---|---|
| 1 | strike | The batter did not swing, The ball was in the strike zone |
| 2 | foul | the batter hit the ball, the ball into the stands, it landed in foul territory |
| 3 | ball | The hitter didn't swing, The ball was outside the strike zone |
| 4 | out | the batter hit the ball, it was caught by the fielder |

**Example** (After Grouping Phase). *We show the grouped concepts for this example corpus*

| Concept-index, $i$ | Concept groups |
|---|---|
| 1 | [The batter did not swing, The hitter didn't swing] |
| 2 | [the batter hit the ball, the batter hit the ball] |
| 3 | [The ball was in the strike zone] |
| 4 | [The ball into the stands] |
| 5 | [it landed in foul territory] |
| 6 | [The ball was outside the strike zone] |
| 7 | [it was caught by the fielder] |

**Example** (After Pruning Phase). *The concept 'the ball into the stands' of Concept-Index 4 from previous table was not contributing much and hence was pruned.*

| Concept-index, $i$ | Concept groups |
|---|---|
| 1 | [The batter did not swing, The hitter didn't swing] |
| 2 | [the batter hit the ball, the batter hit the ball] |
| 3 | [The ball was in the strike zone] |
| 4 | [it landed in foul territory] |
| 5 | [The ball was outside the strike zone] |
| 6 | [it was caught by the fielder] |

**Example** (After Vectorization). *After pruning, each sample is mapped to their corresponding concept vector. The value of concept vector at index $i$ is 1 if $e_n$ has the concept with index $i$, else, it's 0. The Matrix containing all the concept vectors is called the Concept Matric, $C$*

| id, $n$ | label, $l_n$ | Concept Vector, $\mathbf{c}_n$ |
|---|---|---|
| 1 | strike | [1,0,1,0,0,0] |
| 2 | foul | [0,1,0,1,0,0] |
| 3 | ball | [1,0,0,0,1,0] |
| 4 | out | [0,1,0,0,0,1] |

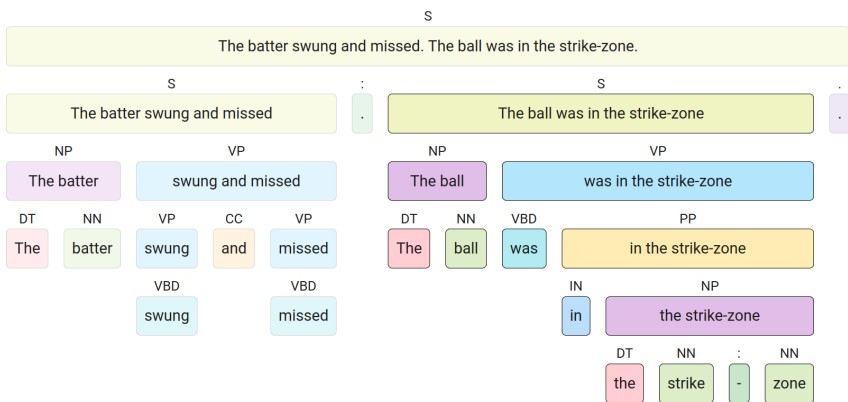

Figure 6: Constituency tree for explanation $e_1$ from Example A.1

## A.2 EXAMPLE CONSTITUENCY TREE

The explanation $n = 1$ from Example A.1 is decomposed into the constituency tree shown in Figure 6. The parser gives a hierarchy of constituents. Our method traverses through this tree and selects the constituents that satisfy the rules discussed in Table 1. It is important to note that rules can be added or deleted at this step based on the requirements.

## A.3 META-DISTANCE FOR LABEL BASED PROXIMITY

At the end of the **Completion Phase** we define a count for each raw concept, $\kappa_i \in \widetilde{\mathcal{K}}$, given by $M_i$. And for each label category $l \in \mathcal{L}$, we define a label count for raw concept $\kappa_i$ as $m_{il}$ where $i$ is the index of the concept. These count the presence of raw concepts explanations, and $\sum_{l \in \mathcal{L}} m_{il} = M_i$. Finally we group together raw concept $\kappa_i$'s label counts into a label count vector $\mathbf{m}_i = [m_{i1}, \ldots, m_{i|\mathcal{L}|}]$

Now we describe the meta-metric $d_{\text{label}}$ used in the **Grouping** phase more formally and provide some intuition behind its construction. Consider that we have two raw concepts $\kappa_i, \kappa_j \in \widetilde{\mathcal{K}}$ and label count vectors $\mathbf{m}_i$ and $\mathbf{m}_j$. We next assume that vector $\mathbf{m}_i$ constitutes $M_i$ i.i.d. draws from a categorical distribution with unknown parameters $\boldsymbol{\mu}_i = (\mu_{il})_{l=1}^{|\mathcal{L}|}$, where $\mu_{il}$ is the probability that a randomly selected occurrence of raw concept $i$ belongs to an entry in the explanation corpus with label category $l$. Our label distance $d_{\text{label}}$ is the *evidence ratio* between the count vectors, $\mathbf{m}_i$ and $\mathbf{m}_j$, being drawn from independent categorical distributions (model $\mathcal{M}_{\text{indp}}$) versus them being drawn from the same distribution (model $\mathcal{M}_{\text{comb}}$). More precisely,

$$d_{\text{label}}(\mathbf{m}_i, \mathbf{m}_j) = \frac{p(\mathbf{m}_i, \mathbf{m}_j | \mathcal{M}_{\text{indp}})}{p(\mathbf{m}_i, \mathbf{m}_j | \mathcal{M}_{\text{comb}})}$$

Note that this is not a true distance between count vectors as two identical count vectors do not have a distance of zero. Nonetheless, it satisfies the other requirements of a metric: non-negativity, symmetry and the triangle inequality, and two vectors that are more (less) likely to come from the same multinomial will have a distance less (more) than 1.

To evaluate the label distance we must calculate the evidence for various categorical samples given the model $p(\mathbf{m}|\boldsymbol{\mu}, M)$. For simplicity, we assume total count $M$ is known and define a Dirichlet prior $p(\boldsymbol{\mu}|\alpha\mathbf{1})$ where $\mathbf{1}$ is the vector of all 1s (this makes the simplifying assumption that the prior is symmetric). The evidence for $\mathbf{m}$ is then:

$$p(\mathbf{m}|\alpha) = \int p(\mathbf{m}|\boldsymbol{\mu}, M) p(\boldsymbol{\mu}|\alpha\mathbf{1}) d\boldsymbol{\mu}$$

The label meta-metric is the evidence ratio given by:

$$d_{\text{label}}(\mathbf{m}_i, \mathbf{m}_j) = \frac{p(\mathbf{m}_i|\alpha) p(\mathbf{m}_j|\alpha)}{p(\mathbf{m}_i + \mathbf{m}_j|\alpha)}$$

For computational efficiency (and since it did not appear to affect results measurably) we use an approximation for $p(\mathbf{m}|\alpha)$ in our calculation of $d_l$.

We first evaluate the expected parameter of the posterior distribution given count vector $\mathbf{m}$, namely

$$\tilde{\boldsymbol{\mu}} = \mathbb{E}[\boldsymbol{\mu}|\alpha, \mathbf{m}]$$

then evaluate the evidence for $\mathbf{m}$ conditioned on $\tilde{\boldsymbol{\mu}}$, i.e.

$$p(\mathbf{m}|\tilde{\boldsymbol{\mu}}) = \prod_{k=1}^{K} \left( \frac{m_{il} + \alpha}{M_i + K\alpha} \right)^{m_{il}}$$

We then calculate $\log d_{\text{label}}(\mathbf{m}_i, \mathbf{m}_j)$ and exponentiate to improve precision. After grouping, the raw concept within a cluster with highest frequency is identified as the representative concept of that cluster.

## A.4 LANGUAGE MODELS

### A.4.1 CONCEPT EXTRACTION

After obtaining the free form textual explanations for both, we first cleaned them by removing explanations associated with corrupted video files and the videos which were labelled incorrectly. We then considered three different Spacy's pretrained constituency parsers: $en\_core\_web\_lg$, $en\_core\_web\_md$, $en\_core\_web\_sm$ to parse the explanations and extract raw concepts based on the rules discussed in section 3.1. We found that, the parser $en\_core\_web\_lg$ was more accurate in identifying the constituents and resulted in better concept extraction.

### A.4.2 CONCEPT GROUPING

For text distance we embedded the raw concepts with a sentence encoder and we experimented with two models: `paraphrase-distilroberta-base-v1` (*distil*) Sanh et al. (2019) and `stsb-roberta-base` (*stsb*) Reimers & Gurevych (2019) both using the `sentence_transformer` python library, and evaluated a variety of distance metrics within the resulting 768 dimensional space, including: Chebyshev (infinity norm), manhattan, Euclidean and cosine distances.

And to cluster the semantically similar concepts together using agglomerative clustering Müllner (2011), we evaluated a variety of distance metrics within the resulting 768-dimensional space, including: Our proposed meta-distance metric, Chebyshev (infinity norm), manhattan, Euclidean and cosine distances. To select hyperparameters, including: choice of sentence embedding model, distance metric for sentence embeddings, prior $\alpha$ for label distance, and relative importance factor $\lambda$ we performed a grid search and selected the values that resulted in well-formed clusters. Based on our experiments (Provided in supplementary materials), we found that *stsb* encoder with our proposed meta-distance metric resulted in the best grouping of concepts. Note: $d_{text}$ can be either cosine or manhattan distance as they gave similar clusters.

After clustering, we set the frequency occurrence threshold of 3 and removed the rare concept groups which occurred less than this threshold. Then, pruning was done using 90% of mutual information score as discussed in section 3.1 which resulted in 80 significant concepts for our MLB-V2E dataset and 62 concepts for MSR-V2E dataset as shown in Table 2 and Figure 7. Therefore, each video was associated with a binary concept vector of shape $[1 \times k]$ where k is the number of concepts, indicating the presence and absence of each concept.

## A.5 THE CLASSIFICATION MODELS

We considered three different *feature extractors* : Resnet 50v2 He et al. (2016), Resnet 101v2 He et al. (2016) and InceptionV3 Szegedy et al. (2016) models and pretrained on the Imagenet dataset to extract features from each frame of our video clips. We excluded the final classification layer from these models and did a global maxpool across the width and height such that we get a 2048 size feature vector for every frame. We then concatenate the features together, resulting in a $[2048 \times 360]$ feature matrix for every video where 360 is the number of frames per video.

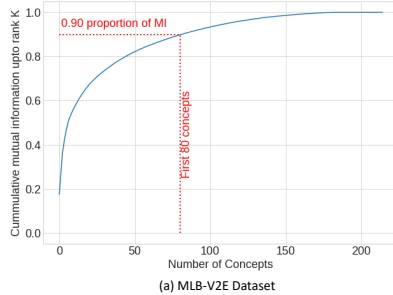 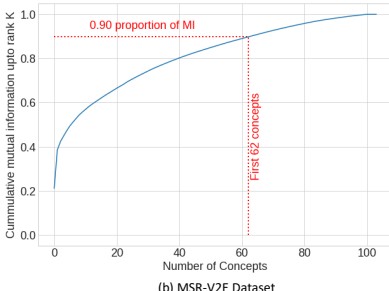

(a) MLB-V2E Dataset    (b) MSR-V2E Dataset

Figure 7: Selecting concepts based on the Mutual Information. (a)MLB-V2E dataset (b)MSR-V2E Dataset

For the *Temporal Layer*, we considered both temporal convolution Lea et al. (2017) and LSTM Hochreiter & Schmidhuber (1997) based architectures which are good at extracting temporal features and found that Temporal CNNs outperformed LSTM by a significant amount. And the *Bottleneck Layer* is a dense layer with $k$ neurons and hence the output is a vector of shape $[1 \times k]$ where $k$ is the number of significant concepts. We introduced an attention layer in the concept-bottleneck model that gives the concept score for each concept. The final fully connected layer is implemented with $L$ neurons (L classes) which predicts the class from the video.

**Model Loss function** Assuming $s$ is the feature vectors obtained from the Feature extractors

$$Loss(L) = \frac{1}{N} \sum_{n=1}^{N} \left( L_{Y_n} + \beta \times L_{C_n} \right) \tag{4}$$

$$= \frac{1}{n} \sum_{i=1}^{n} \left( \beta \sum_{k=1}^{l} \left[ -c_k^i \log(f_\sigma(s_k)) - (1 - c_k \log(1 - f_\sigma(s_k))) \right] - \sum_{j=1}^{m} y_j \log f_S(s_j) \right)$$

$$\text{where} \qquad f_\sigma(s_i) = \frac{1}{1 + e^{-s_i}} \qquad \text{and} \qquad f_S(s_i) = \frac{e^{s_i}}{\sum_{j=1}^{m} e^{s_j}} \qquad \text{and} \qquad \beta > 0$$

## A.6 SELECTING CONCEPTS BASED ON THE MUTUAL INFORMATION(MI)

Figure 7 shows the plot between cumulative MI and the number of concepts after pruning. As discussed in Section 5 the sweet spot for the number of concepts corresponded to 90% of the cumulative MI beyond which there was no gain in classification performance as we increased the number of concepts.

## A.7 PERFORMANCE OF MODELS

Table 5 shows the performance of all the models with different feature extractors. Each model was trained thrice and the mean and standard deviations are reported. We find that models with Inception V3 as the feature extractor performed the best. Adding attention mechanism greatly improved the performance of concepts prediction and also achieved higher accuracies than the concept bottleneck models without attention.

## A.8 THE RELATIONSHIP BETWEEN CONCEPTS AND THE CLASSIFICATION TASK

To understand the relationship between the extracted the concepts and the task classification, we compared the performance of the Concept Bottleneck models with a) MLP classifier b)Linear Classifier as the final classification layers. The results showed that there was approximately 2% drop in classification performance when using a Linear Classifier instead of an MLP. This indicates that, the classification task is not a simple linear combination of the extracted concepts and the composition of concepts is important.

| Dataset | Feature Extractor | Model Type | Task Classification | | Concepts |
|---------|-------------------|------------|---------------------|-----|----------|
| | | | Accuracy(%) | F1-score | AUC |
| MLB-V2E | Resnet 50V2 | Standard | $67.92 \pm 0.78$ | $0.68 \pm 0.003$ | - |
| | | Bottleneck | $67.83 \pm 0.74$ | $0.68 \pm 0.001$ | $0.85 \pm 0.005$ |
| | | Bottleneck + Attn. | $67.96 \pm 0.65$ | $0.68 \pm 0.002$ | $0.88 \pm 0.002$ |
| | Resnet 101V2 | Standard | $68.18 \pm 0.88$ | $0.68 \pm 0.005$ | - |
| | | Bottleneck | $68.01 \pm 1.02$ | $0.68 \pm 0.013$ | $0.85 \pm 0.004$ |
| | | Bottleneck + Attn. | $68.26 \pm 1.12$ | $0.68 \pm 0.009$ | $0.88 \pm 0.000$ |
| | Inception V3 | Standard | $68.46 \pm 1.27$ | $0.68 \pm 0.011$ | - |
| | | Bottleneck | $68.16 \pm 1.12$ | $0.68 \pm 0.004$ | $0.85 \pm 0.003$ |
| | | Bottleneck + Attn. | $68.38 \pm 1.34$ | $0.68 \pm 0.004$ | $0.88 \pm 0.001$ |
| MSR-V2E | Resnet 50V2 | Standard | $61.52 \pm 1.20$ | $0.59 \pm 0.008$ | - |
| | | Bottleneck | $61.23 \pm 1.71$ | $0.59 \pm 0.008$ | $0.82 \pm 0.009$ |
| | | Bottleneck + Attn | $61.28 \pm 1.54$ | $0.59 \pm 0.007$ | $0.86 \pm 0.004$ |
| | Resnet 101V2 | Standard | $61.56 \pm 1.31$ | $0.60 \pm 0.008$ | - |
| | | Bottleneck | $61.38 \pm 1.24$ | $0.60 \pm 0.008$ | $0.83 \pm 0.006$ |
| | | Bottleneck + Attn. | $61.44 \pm 1.32$ | $0.60 \pm 0.009$ | $0.86 \pm 0.003$ |
| | Inception V3 | Standard | $61.79 \pm 1.42$ | $0.60 \pm 0.012$ | - |
| | | Bottleneck | $61.42 \pm 1.18$ | $0.60 \pm 0.013$ | $0.83 \pm 0.006$ |
| | | Bottleneck + Attn. | $61.68 \pm 1.23$ | $0.60 \pm 0.009$ | $0.86 \pm 0.004$ |

Table 5: Performance of Models

| Dataset | Accuracy (%) | | Difference (%) |
|---------|--------------|------------------|----------------|
| | MLP | Linear Classifier | |
| MLB-V2E | 68.38 | 66.94 | -1.44 |
| MSR-V2E | 61.68 | 60.02 | -1.66 |

Table 6: Performance of MLP classifier vs Linear Classifier

| Dataset | Task Accuracy(%) | Task F-1 score |
|---------|------------------|----------------|
| MLB-V2E | 75.68 | 0.7585 |
| MSR-V2E | 65.23 | 0.6422 |

Table 7: Performance of MLP classifier trained only on concepts (without videos)

## A.9    A CLASSIFIER MODEL TRAINED ONLY ON CONCEPTS

The MLP model trained on only the concepts (without using video information) achieves a higher accuracy (7% for MLB-V2E and 4% for MSR-V2E) than the video classification model. This result indicates that the concepts extracted are meaningful for the classification task and there's still an headroom available if the concept bottleneck network was able to predict these initial concepts better.

## A.10    MORE EXPLANATIONS FROM THE PREDICTION MODEL

Here are a few more examples of the prediction by the concept bottleneck model with attention. Figure 8 shows examples from the MLB-V2E dataset and Figure 9 shows examples from the MSR-V2E dataset. The example videos are provided in the supplementary materials.

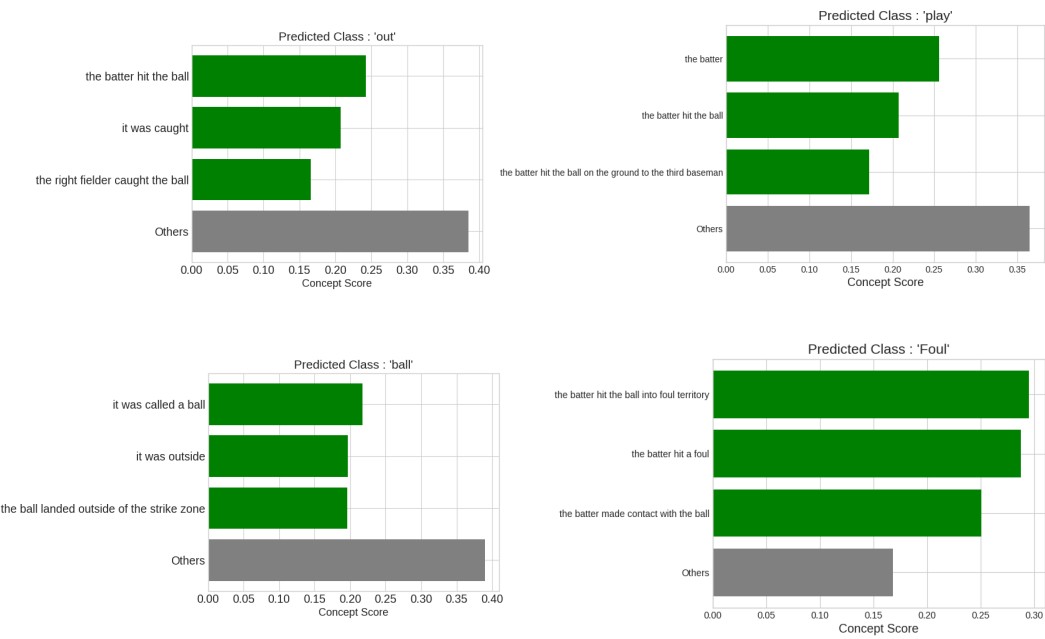

Figure 8: Examples of the model prediction and their corresponding concepts and their importance scores for MLB-V2E dataset

## A.11    MORE DETAILS ON THE DATASETS

Figure 10 shows the number of videos belonging to each category in the MLB-V2E and the MSR-V2E datasets. Though the original MSR-VTT dataset had descriptions of videos, they were general captions and didn't explain any particular class. Since they didn't have classification labels and text-based explanations corresponding to the labels for the videos, we collected the video labels and natural language explanations by crowd-sourcing on Amazon Mechanical Turk. The explanations obtained for these videos and the concepts extracted using *CoDEx* for both the datasets are provided in the supplementary materials. Since the MSR-V2E dataset is imbalanced, we do some weighted oversampling while training to ensure that the models learn to predict all the classes.

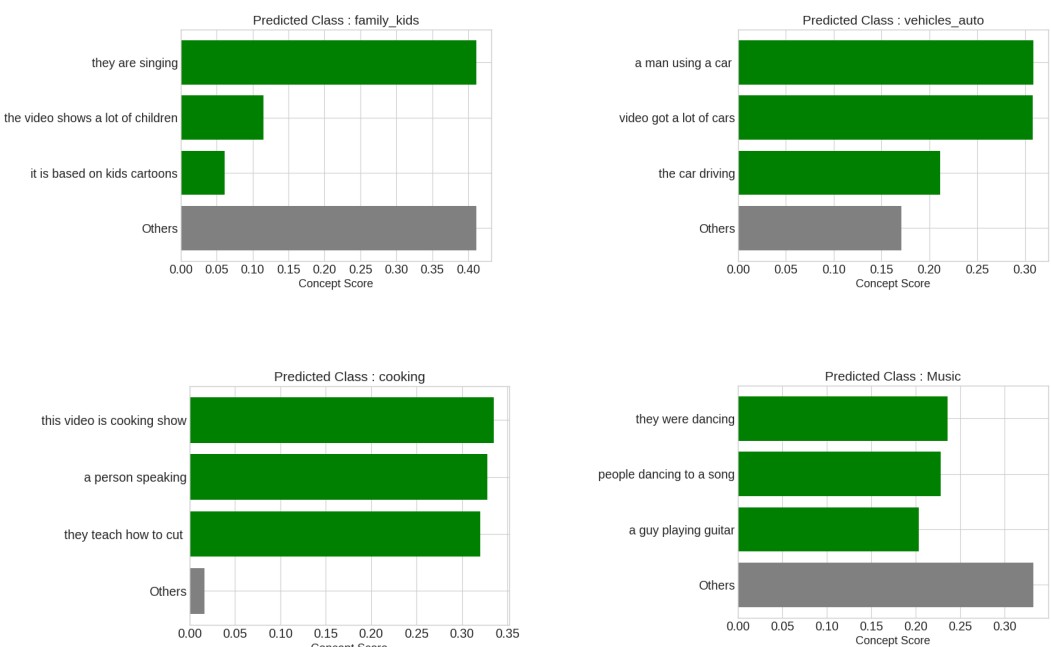

Figure 9: Examples of the model prediction and their corresponding concepts concepts and their importance scores for MSR-V2E dataset

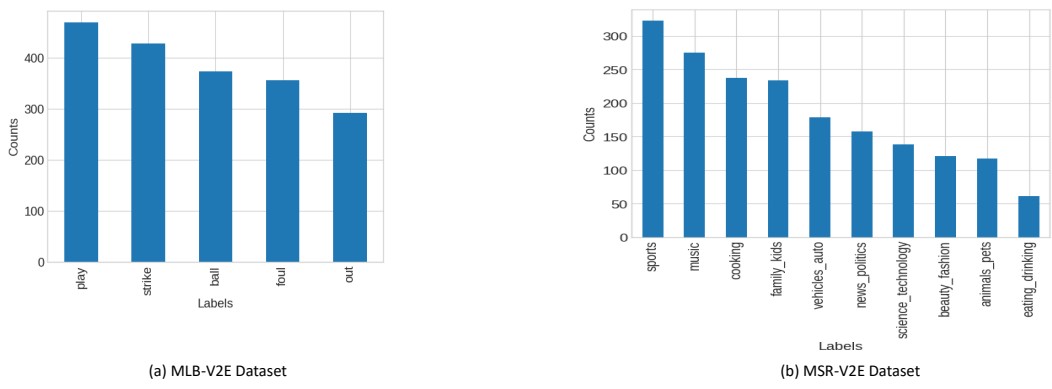

Figure 10: The number of videos belonging to each category on (a) the MLB-V2E dataset and (b) the MSR-V2E dataset

