# OpenReview forum: "Automatic Concept Extraction for Concept Bottleneck-based Video Classification"
_ICLR.cc/2022/Conference — ICLR 2022 Submitted_

### Official Review · Reviewer_LdBL · 2021-10-31

**Correctness:** 3
**Technical Novelty And Significance:** 2
**Empirical Novelty And Significance:** 2
**Recommendation:** 5
**Confidence:** 4

**Main Review:**

## Strengths
1. The ways the method groups and prunes raw concepts are interesting as these sub-modules are aware of associate labels.
1. The experimental results are encouraging. Because the concept bottleneck forces to drop much information in the original video, the performance drop seems inevitable; yet the performance degradation is limited.

## Weaknesses
1. I think the competing approach with this method is the one that uses a set of handcrafted concepts instead of automatically extracted concepts from textual descriptions, while according to the paper, the downside of such handcrafted concepts is that there is no guarantee of having a necessary and sufficient set of concepts. However, this is also the case for textual descriptions, and thus to demonstrate the merit of this method, I think comparison to the approach that uses a set of handcrafted concepts are necessary. One possible comparison is to design such a set of concepts from scratch, and another is to manually extract concepts from textual descriptions.
1. The necessity of pruning is not clear. According to Figs. 3 and 4, the number of concepts (almost) monotonically increases the performance in senses of both MI and classification. I would say that, perhaps, these less useful concepts do not help a lot, but unless they are harmless (for example, redundant concepts may still increase the chance of correct classification, while if a certain concept is completely useless, the learned FC layer will ignore it), there is no need to remove them. It would be helpful to have some more discussion on the necessity of pruning.


**Summary Of The Paper:**

This paper presents a method for video classification, which uses textual descriptions of each training video to automatically extract concepts and uses these concepts for classification. For extracting the concepts, the method identifies some phrases in each description. The extracted phrases are grouped based on their semantic similarity and associated labels. Then they are pruned so that only informative phrases are extracted as concepts. For inference, these concepts are identified in a video and based on these identified concepts, an FC layer classifies a video into one of the target classes. The method is evaluated with two datasets MLB-Youtube video activity classification dataset and MSR VTT dataset with extending them with textual descriptions, showing that the concept bottleneck does not impact a lot to the classification performance. The human study demonstrates that identified concepts by their method is better than randomly selected/frequent concepts.

**Summary Of The Review:**

I think the paper is well-written, but the merit of the method is not very clear. Since textual descriptions may require a similar burden to handcrafting a set of concepts, the proposed method may just add noises in the set of concepts. Regarding this point, the merit of the method should be experimentally verified (w1). Also, the necessity of pruning is not very clear (w2), which I guess involves the first contribution of the paper. I think the paper will be stronger if these concerns are resolved.

---

> ### Author Response · Authors · 2021-11-15
> **Response from authors**
>
> Thank you for your review and valuable feedback. We have listed the concerns raised in the reviews and provided our response in the general comment. Please let us know if you have any comments or suggestions.

---

> > ### Comment · Reviewer_LdBL · 2021-11-29
> > **Thanks for clarifiation**
> >
> > I appreciate the authors' effort to address the reviewers' concerns.
> >
> > However, I'm not very convinced about the first point of the weaknesses, as the method still requires expert annotation by text, and I have to keep my score.

---

### Official Review · Reviewer_9TvQ · 2021-11-04

**Correctness:** 3
**Technical Novelty And Significance:** 2
**Empirical Novelty And Significance:** 2
**Recommendation:** 5
**Confidence:** 3

**Main Review:**

Strength:
1) CoDEx proposed in this paper provides a more comprehensive understanding of the video feature by extracting concepts from the explanation of the labels and constructing a higher-dimensional concept space.
2) The pipeline can be easily applied to any supervised video classification architecture.

Weakness:
1) One of the motivations of the CoDEx is to get concepts without domain experts, but only simple cases are reported in the results. In a simpler domain, the label will be highly correlative to the concepts, so the requirement of experts is low. However, in a more complicated domain, the requirement of experts will become higher. Whether the explanation will still be reasonable is not reported.
2) This work is more like a "label decomposition" based on natural language explanations and has low relevance to video modeling.
3) The annotation quality of explanations is important since the extracted concepts are from them. However, only two datasets with 5 and 10 classes are reported in this paper. It seems that the annotation cost of this pipeline is high in even normal-size datasets.
4) In the extraction phase of CoDEx, a fixed set of rules is used to extract "raw concepts". The motivation for using sentences as the concept is not clear. Will word-level concepts work?

Minor: Both "black-box" and "blackbox" are shown in the paper, which is inconsistent.

**Summary Of The Paper:**

 This paper proposed an automatic concept discovery and extraction module based on the label explanations and an end-to-end concept bottleneck model for video classification. The result on two constructed datasets (MLB-V2E and MSR-V2E) shows that the architecture can get explainable results without harming the classification performance.


**Summary Of The Review:**

This paper proposed an end-to-end pipeline to train an explainable video classification model. The paper is targeted to study three interesting research questions, and I particularly liked the discussion section at the end of the paper. However, the experiments lack in some aspects resulting in a less convincing story.


######## Post-Rebuttal ########

I appreciate the authors made a good effort to improve their paper based on review comments. However, I still find that the contribution is weak -- experiments are done on only two video datasets with limited classes but given that the method itself is generic. Therefore, I stick with my original rating.

---

> ### Author Response · Authors · 2021-11-15
> **Response from authors**
>
> Thank you for your review and valuable feedback. We have listed the concerns raised in the reviews and provided our response in the general comment. Please let us know if you have any comments or suggestions.

---

### Official Review · Reviewer_uphf · 2021-11-08

**Correctness:** 4
**Technical Novelty And Significance:** 2
**Empirical Novelty And Significance:** 2
**Recommendation:** 5
**Confidence:** 3

**Main Review:**

The authors address the timely problem of interpretable video classification. The paper is well-written, and the authors have also described their method, and data collection process, thoroughly and clearly. And I also appreciate the discussion of future work and weaknesses of the current approach.

However, I do feel that the main contribution of the paper -- to generate concepts automatically from natural language descriptions provided by the annotators -- is quite heuristic (combination of NLP post-processing and clustering) and not generalisable across different datasets.

It is not clear to me how the proposed method can be extended to more common video datasets. For example, on Kinetics or Something-Something, the action classes are fairly "atomic", and it is not clear to me what natural language explanations there could for these kind of actions beyond the class label itself. In other words, the "concepts" that end up being extracted in this paper, are similar to the class labels themselves in other datasets. And so it is not clear to me how the proposed method can develop interpretable models for such datasets.

I think the paper would also be stronger if the authors compared the annotation effort required to generate the natural language descriptions used in this paper, to the annotation time/cost to collect the original dataset in the first place, compared to the annotation effort required for the manually-defined set of concepts in previous work. Although the proposed concept extraction method is automatic, it still requires significant annotations in the form of natural language descriptions for each video in the dataset. And so the annotation time and effort of this needs to be contextualised.

The paper would also be more convincing to me if the authors augmented existing, popular video classification datasets (ie Kinetics, Something-Something, Epic Kitchens etc) with descriptions and developed interpretable models for such datasets. Not only are these datasets more common in the community, but I believe that the vocabulary of these datasets are more challenging for the proposed method, compared to for example, the baseball dataset used in this work (since sports videos consist of well-defined actions which are explained by the rules of the sport.)

The proposed attention mechanism helps to improve the concept prediction accuracy, but not at all on the task classification accuracy. It would also be interesting to see how the fully-connected subnetwork, operating on "oracle" concept scores (ie the concept ground truth, rather than the concepts predicted by the network before it) performs. This would show the headroom available if the network was able to predict these initial concepts better.

**Summary Of The Paper:**

The authors address the task of concept-based video classification: The resulting model is interpretable, as it first predicts a set of binary "concepts", and these concept-features alone are then used for producing the final classification. The main novelty of this work is to propose a method for automatically generating these concepts from natural language descriptions (hence avoiding the need for expert annotations). The authors produce two new datasets of roughly 2000 videos each to validate their method. They also show that their concept-based model performs as well as a standard video classification model on these datasets (ie interpretability does not come at the cost of predictive accuracy).

**Summary Of The Review:**

In summary, the paper is well-written and addresses an important problem of interpretable video classification models. The proposed method does not rely on manual concept labelling, but rather extracts these concepts automatically from natural language descriptions provided by dataset annotators. My concern is that the proposed method does not seem general to me, and I feel that it only works for the specific datasets that the authors have chosen. And although the concept extraction process is automatic, the method requires obtaining natural language descriptions for each video in the dataset, and the annotation cost of this is not discussed at all in the paper.

---

> ### Author Response · Authors · 2021-11-15
> **Response from authors**
>
> Thank you for your review and valuable feedback. We have listed the concerns raised in the reviews and provided our response in the general comment. Please let us know if you have any comments or suggestions.

---

> > ### Comment · Reviewer_uphf · 2021-11-29
> > **After reading rebuttal**
> >
> > Thank you for your rebuttal, and responding to all reviewers' comments in detail.
> >
> > I am, however, sticking with my original rating. In my opinion, the method involves multiple heuristics, and is not generalisable to different datasets (ie, it is only suitable for the types of "complex" actions which can be narrated by a human annotator). Also, for an "automatic" method, it still requires a lot of human annotation in the form of narrations, and the cost of this has not been quantified at all.

---

### Official Review · Reviewer_zzha · 2021-11-08

**Correctness:** 3
**Technical Novelty And Significance:** 2
**Empirical Novelty And Significance:** 3
**Recommendation:** 6
**Confidence:** 3

**Main Review:**

Here are the pros and cons of the paper:

Pros:
1. This paper presents a quite interesting method which automatically selects the concepts as bottleneck for explanation. This could benefit other interpretability models which have textual descriptions available.
2. The authors performed a good amount of illustration on the intermediate results after each phase
3. The proposed datasets could also benefit the related fields/applications

Cons:
1. It seems that there's a lack of ablation studies on each phase and a lack of some decent baseline models for selecting the key concepts. One simple baseline might be, training an MLP together with the information bottleneck model and selecting the ones with high importance. For each phase, it's not quite clear to me how important they are.
2. It looks like the CoDex also needs some human efforts to tune and intervene the process. Is there any way to empirically prove that they are general enough to process various texts and can obtain similar results with humans?
3. The authors mentioned that this system is for videos, but it looks like the whole process is quite general, and there is not much correlation with the mentioned issues such as the time span?

**Summary Of The Paper:**

This paper works on the interpretability of video understanding problem. With a set of textual descriptions, the authors propose a pipeline called CoDex to extract the key concepts for explaining the classification, in contrast to previous methods which use the predefined classes. The CoDex method contains clearning, extraction, completion, grouping, pruning and vectorization phases to obtain the final concept matrix. The trained final concept bottleneck model can obtain interpretable explanations aligned with human servey, while retaining the performance

**Summary Of The Review:**

I think this paper overall contributes an interesting pipeline for automatically extracting relevant concepts for the concept bottleneck model. Several concerns are listed above.

-- after rebuttal --
Thanks for providing the rebuttal and explanations. I've read the authors' responses and other reviews. Overall I still feel the paper is addressing an interesting problem, but agree that the method might limit its applicability. I'll keep my rating (borderline-ish).

---

> ### Author Response · Authors · 2021-11-15
> **Response from authors**
>
> Thank you for your review and valuable feedback. We have listed the concerns raised in the reviews and provided our response in the general comment. Please let us know if you have any comments or suggestions.

---

### Author Response · Authors · 2021-11-15
**Response to Reviewers**

We thank all the reviewers for their thoughtful reviews. The following response seeks to address the reviewers&#39; concerns:

**Annotation Effort:**

_R3 &quot;It seems that the annotation cost of this pipeline is high&quot;_

_R2&quot; the annotation time and effort of this needs to be contextualised&quot;_

We do agree that there&#39;s effort involved in annotating the dataset with natural language explanations. But we should note that the prior work requires identifying the necessary set of concepts that have to be annotated. Therefore, identifying the set of important concepts will need a separate study, and then annotating those concepts would require another study resulting in two independent studies, which makes it more time-consuming and costly. Further, if the vocabulary of concepts is large, the annotation process would be inconvenient for the user. In our case, we significantly reduce the effort to identify and annotate the concepts with natural language explanations in a single user study. Moreover, natural language explanations can express richer compositions of concepts rather than simply identifying the presence or absence of an individual concept. Therefore, we did not conduct a user study on the differences in annotation effort as it seemed self-evident that performing two separate studies and using a large number of concepts is more expensive and cumbersome.

Another way to reduce the annotation effort is to use the readily available commentaries (usually available for sports videos) for our purposes. However, we would need to identify and synchronize appropriate commentaries with video clips, and care needs to be taken over how these commentaries are treated as explanations of labels. Thus, we will add a small discussion on how future work may leverage existing expert commentary as natural language explanations.

**Heuristic Nature and generalizability Concept Extraction:**

_R1 &quot; prove that they are general enough to process various texts &quot;_

_R2 &quot;generate concepts automatically from natural language descriptions provided by the annotators -- is quite heuristic and not generalisable across different datasets &quot;_

The pipeline has some heuristic aspects, but these heuristics are relatively general and based on sentence structure. We aim to extract text fragments and derive simpler sentences from the more complex language used in the annotations (not extract the sentences as is) that correspond to concepts that are relevant to the classification. Note that this concept extraction process is based on the structure of the language used. Our paper showed that the language could be about baseball plays (MLB-V2E) or general video categorization across a wide variety of genres (MSR-V2E); these two domains are arguably very different and utilize very different domain-specific vocabulary. However, in both cases, the natural language descriptions are written in English, and the sentence structures (as discovered by NLP parsers) share sufficient commonalities for the method to work well in both cases.

**Comparison with a baseline concept extraction method:**

_R3 &quot;The motivation for using sentences as the concept is not clear. Will word-level concepts work?&quot;_

_R4 &quot; comparison to the approach that uses a set of handcrafted concepts are necessary&quot;_

We argue that words alone are not sufficient, and we will be evaluating that by replacing concept matrices with unigram matrices (word-level concepts) that will act as a baseline concept extraction. We will include the results and discussion in our revised version of the paper within next week. (Please see the end of the comments on proposed experiments/ablation studies).

**Importance of each phase:**

_R1 &quot;It seems that there&#39;s a lack of ablation studies on each phase&quot;_

_R4 &quot;The necessity of pruning&quot;_

We agree that having an ablation study, especially for Grouping and Pruning, will show their importance and improve the quality of our paper. Hence, we will be including the experiments and results in a revised version which we will provide within the next week. (Please see the end of the comments on proposed experiments/ablation studies).

To address the necessity of pruning in particular, we aim for an explainable model, and a substantial reduction in low information concepts facilitates that. Although these concepts may be harmless from the perspective of classification performance, they can impact the concept accuracy and explainability value of the result. Further, having fewer concepts significantly reduces the size of the classification model as well. We will include these points in the discussion.

---

> ### Author Response · Authors · 2021-11-15
> **Response to Reviewers - Continuation**
>
> **The Application of Concept-Bottleneck based models on Popular Datasets:**
>
> _R2 &quot;can be extended to more common video datasets? For example, on Kinetics or Something-Something, the action classes are fairly &#39;atomic&#39; &quot;_
>
> This method of using CBM is only applicable to those tasks where an explanation by concepts is applicable. This relates to the intended use case of our approach, which we see as explaining predictions on non-atomic classes. As such, these more complex predictions will be explained in terms of more atomic concepts. We see our method as valuable in areas such as human decision support with video surveillance or video retrieval from large datasets based on higher-level concepts. In those domains, it is the decomposition of higher-level events into more atomic events that is key. The Something-something dataset, for example, doesn&#39;t represent a database of videos that can be decomposed into higher-level concepts; instead, it is specifically constructed as a dataset for training and benchmarking methods that perform lower level/atomic activity recognition. We will include this in our discussion and clarify that CBM is only applicable to complex tasks.
>
> **The necessity of domain experts:**
>
> _R3 &quot;However, in a more complicated domain, the requirement of experts will become higher.&quot;_
>
> In previous methods, the domain experts had to come up with a list of all necessary concepts for classification before they could be annotated. That is a highly challenging task. In our approach, by using natural language explanations, we removed the need for domain experts to identify the necessary and sufficient set of concepts. However, we still need some form of expertise to provide natural language explanations. We will clarify this point throughout the paper: we are not removing the need for experts altogether but significantly reducing the concept identification and annotation procedures. Moreover, natural language explanations provide the ability to express richer compositions of concepts rather than simply identifying the presence or absence of an individual concept.
>
> **Headroom for improvement:**
>
> _R2 &quot; It would also be interesting to see how the fully-connected subnetwork, operating on concept scores (ie the concept ground truth, rather than the concepts predicted by the network before it) performs &quot;_
>
> We will train an MLP on ground-truth concepts to identify the headroom available if the network was able to predict these initial concepts better. We will include a section in the appendix to add this evaluation and discussion in our revised version of our paper.
>
> **Why Video Classification?:**
>
> R1 &quot;The authors mentioned that this system is for videos, but it looks like the whole process is quite general,&quot;
>
> As R3 correctly mentioned in one of their statements, &quot;in a simpler domain, the label will be highly correlative to the concepts&quot;. Therefore, we considered the more complex video domain where there is a particular need to compose and relate concepts to one another, and the benefits of our approach are observable.
>
> **Proposed Experiments that will be included in the revision:**
>
> To address explicit points made by the reviewers regarding the empirical studies, we will run the following studies and include the results in a revised submission before the end of the review period:
>
> 1. We will perform an ablation study removing the raw concept extraction and replacing it with unigram (word-level) concepts: We will keep everything else the same, and we expect performance to drop. This addresses the concern that a) the NLP raw concept extraction is arbitrary heuristic and b) that concepts must be something more complex than words.
> 2. We will perform an ablation study removing the grouping part and pruning part of the NLP pipeline. This directly addresses the need for an ablation study with key features of the pipeline [R1, R4].
> 3. We will also perform an ablation study replacing the final MLP with Logistic regression for video classification. If performance drops without a hidden layer, this means that the composition of concepts is important and is not a simple linear combination of the extracted concepts.
> 4. We will train a fully-connected network operating on concepts to show the headroom available if the network was able to predict these initial concepts better as suggested by R2.

---

### Author Response · Authors · 2021-11-22
**Revised Version**

We thank the reviewers again for their constructive reviews.
Here is the summary of the list of changes incorporated in the revised version:
- Included a discussion on the annotation effort and benefits of our annotation method in the 'Discussion' section of the paper
- Expanded the Results section with the results from the experiments/ Ablation studies (proposed in the response) that indicate the importance of each step in the pipeline and the reason for choosing phrases over word-level concepts.
- Added a section to the Appendix that talks about the headroom available if the concept bottleneck model was able to predict these initial concepts better as suggested by Reviewer 2
- Made sure the spelling of 'black-box' is consistent

---

### Decision · Program_Chairs · 2022-01-20

**Decision:**

Reject

**Comment:**

The authors consider the task of interpretable video classification. First, a set of binary “concepts'' is predicted, and these concept features are then used for classifying a video. The set itself is automatically generated from natural language descriptions, instead of relying on expert annotations. The authors collect two datasets to validate the proposed approach and show that the model can match the performance of a standard video classification model, while being interpretable.

The reviewers felt that the paper was well written and that the method and empirical results were clearly outlined. They also appreciated the empirical results whereby interpretability doesn’t necessarily come at the expense of accuracy and consider interpretability as a desirable property. The main reason for the borderline results is the heuristic nature of the proposed automatic concept labeling and the empirical evaluation against alternative baselines. In particular, one needs to **show that the proposed method generalises to other datasets**. Secondly, one of the main contributions, namely the automatic **concept extraction, still ends up requiring human annotation in the form of narrations**, and this cost should be quantified and contextualised.

I suggest the authors address these points and resubmit.